# Hyperbaric oxygen therapy in alleviating cerebral ischemia-reperfusion injury via the BMP6/Smad-hepcidin pathway

Lan-Zhao Wang[1], Ji-Hong Zhang[2], Ji-Min Shi[2], Xiu-Ju Li[3]*

1 Center of Emergency and Critical Medicine, Jinshan Hospital of Fudan University; Research Center for Chemical Injury, Emergency and Critical Medicine of Fudan University; Key Laboratory of Chemical Injury, Emergency and Critical Medicine of Shanghai Municipal Health Commission, Shanghai, China, 2 Department of Clinical Medical Research Center, Jinshan Hospital of Fudan University Shanghai, China, 3 Department of Occupational Disease Prevention and Control, Jinshan Hospital of Fudan University; Key Laboratory of Chemical Injury, Emergency and Critical Medicine of Shanghai Municipal Health Commission, Shanghai, China

* lixiuju2008@126.com

## Abstract

Cerebral ischemia-reperfusion injury (CCI) is a cause of neurological damage. Hyperbaric oxygen therapy (HBOT) can improve recovery in CCI in relation to iron metabolism and ferroptosis, but the precise mechanisms remain unclear. This study aims to explore the neuroprotective effects of HBOT in CCI and its regulator of iron homeostasis via BMP6/Smad-Hepcidin signaling pathway. Male Wistar rats were divided into Control (CT), Ischemia-Reperfusion (GM), Ischemia-Reperfusion + Normobaric Hyperoxia (NH), and Ischemia-Reperfusion + HBOT (HO) groups. The CCI model was induced by four-vessel occlusion. HBOT was administered at 2.5 ATA for 120 minutes daily for 5 days. Neurological function was assessed using the modified neurological severity score, light-dark box, and Morris Water Maze test. Histopathological analysis, transmission electron microscopy, Nissl and TUNEL staining, oxidative stress markers, Western blotting and qPCR were used to assess neuronal damage, mitochondrial integrity, necrosis, apoptosis, oxidative stress, iron metabolism and BMP6/Smad-Hepcidin mRNA expression and protein concentrations. HBOT significantly improved neurological function, reduced neuronal damage, and preserved mitochondrial integrity compared to untreated animals. Oxidative stress markers, including malondialdehyde and antioxidant enzyme activities were significantly restored. HBOT also downregulated the BMP6/Smad-Hepcidin pathway, leading to decreased hepcidin levels. Western blot and qPCR analysis confirmed the suppression of ferroptosis-related markers in the HBOT group. HBOT significantly reduces neurological deficits, neuronal damage, and oxidative stress in CCI injury. Its neuroprotective effects are likely mediated by the regulation of the BMP6/Smad-Hepcidin pathway and the suppression of ferroptosis. These findings suggest that HBOT is a promising therapeutic strategy for treating CCI.

**Data availability statement:** The data underlying the results presented in the study are available from: https://github.com/jsyyky/BMP6-Smad-Hepcidin.

**Funding:** This research was funded by the Jinshan Hospital (JYQN-JC-202207) and Jinshan District Science and Technology Commission (2023-WS-22) to investigator Lan-Zhao Wang.The funders had no role in study design, data collection and analysis, decision to publish, or preparation of the manuscript.

**Competing interests:** The authors have declared that no competing interests exist.

## Introduction

Cerebral ischemia-reperfusion injury (CCI) is one of the most critical reasons for severe neurological deficits. CCI leads to interrupted brain circulation, hypoxia and subsequent ischemic damage to brain tissue [1]. The complex pathophysiology of CCI includes mechanisms such as excitotoxicity, oxidative stress, calcium overload, and inflammation [2]. During ischemia, cellular damage is caused by the interruption of the brain's energy supply, while reperfusion further exacerbates injury [3]. This process generates a large quantity of reactive oxygen species (ROS), which damage cell membranes, proteins, and DNA, triggering neuronal death. Additionally, activation of microglia and astrocytes promotes a pro-inflammatory environment that further contributes to neural damage [4]. These combined effects make CCI a significant challenge in clinical management.

Recent studies suggest that ferroptosis plays a critical role in CCI. Ferroptosis is a form of iron-dependent cell death, which is characterized by lipid peroxidation, ROS accumulation, and an over-reliance on iron metabolism. Ferroptosis leads to cellular damage distinct from other forms of cell death such as apoptosis and necrosis [5]. Previous studies showed that the excessive accumulation of ROS leads to the breakdown of cell membranes and ferroptosis [6]. Furthermore, iron dysregulation has been linked to the initiation of ferroptosis in CCI, which highlights the importance of targeting iron metabolism in therapeutic interventions [7]. The BMP6/Smad signaling pathway is a key regulator of iron homeostasis, which has been identified as a potential modulator of ferroptosis [8]. The BMP6/Smad pathway controls the expression of hepcidin, which regulates iron levels by inhibiting iron absorption and promoting iron sequestration in tissues [9].

Hyperbaric oxygen therapy (HBOT) has emerged as a potential therapeutic approach to mitigate the damage caused by CCI. Previous studies have demonstrated that early administration of HBOT significantly enhances neurological recovery and reduces long-term brain damage [10]. HBOT works by increasing oxygen delivery to hypoxic tissues, promoting cellular repair mechanisms, and reducing oxidative stress [11]. A previous study showed that HBOT improves CCI in rats via inhibition of ferroptosis [12]. However, the precise mechanisms by which HBOT alleviates CCI remains unclear.

We hypothesized that HBOT may exert its effect by modulating the BMP6/Smad-Hepcidin pathway. By regulating this pathway, HBOT could potentially stabilize iron metabolism, limit ROS production, and prevent lipid peroxidation, then protect brain cells from ferroptosis. Thus, this study aims to explore the protective mechanisms of HBOT in CCI, with a specific focus on its modulation of the BMP6/Smad-Hepcidin signaling pathway.

## Materials and methods

### Ethical considerations

All procedures were approved by the Laboratory Animal Welfare & Ethics Committee (2024-A041-01) and were conducted in accordance with the guide for the care and

use of laboratory animals. All surgery was performed under sodium pentobarbital anesthesia, and all efforts were made to minimize suffering.

## Animals

Male Wistar rats, weighing between 250–280 g, were obtained from Shanghai Slac Laboratory Animal Company. The animals were housed in a temperature-controlled room (22±2°C) under a 12-hour light/dark cycle with unrestricted access to food and water. Prior to experimentation, the rats were allowed a one-week acclimatization period to minimize stress.

## Experimental groups

The rats were randomly assigned to one of four experimental groups (n=9 per group), namely Control Group (CT): The rats were anesthetized and subjected to sham surgeries without occlusion of the arteries. Global CCI Model Group (GM): The rats underwent the CCI procedure without any treatment. CCI+HBOT Group (HO): These animals underwent the CCI procedure followed by daily HBOT. CCI+Normobaric Hyperoxia Group (NH): These rats underwent the CCI procedure followed by daily Normobaric Hyperoxia.

## Global Cerebral ischemia-reperfusion model

Four-vessel occlusion (4VO) model was employed to induce CCI. Rats were anesthetized using 2–3% isoflurane and placed in the supine position. A midline incision was made in the neck to expose the bilateral common carotid arteries (CCAs) and vertebral arteries. Permanent occlusion of the vertebral arteries was performed using electrocautery through a small dorsal incision at the level of the first cervical vertebra. The bilateral CCAs were then temporarily occluded using microvascular clamps for 30 minutes, inducing cerebral ischemia. Reperfusion was initiated by removing the clamps after the ischemic period. The success of ischemia was confirmed by the following criteria: loss of consciousness, dilated pupils, absence of corneal reflex, and lack of the righting reflex. Reperfusion lasted for 24 hours, after which the animals received therapeutic interventions based on their group assignments.

## Hyperbaric oxygen therapy

The treatment started at 24 h after reperfusion. HBOT was administered in a hyperbaric chamber capable of maintaining a pressure of 2.5 atmospheres absolute (ATA). Rats in the HO group were placed in the chamber with the pressure gradually increased over a 25-minute period to 2.5 ATA. The animals were exposed to 100% oxygen for 70 minutes at this pressure, followed by a gradual depressurization phase lasting 25 minutes. HBOT was administered once daily for five consecutive days. In contrast, rats in the NH group were treated with 100% oxygen at atmospheric pressure (1 ATA) following the same time schedule.

## Neurological assessment

All the neurological assessment started on the 6th day after reperfusion using the modified neurological severity score (mNSS), light-dark box test (LDB) and Morris Water Maze (MWM). The mNSS scale ranges from 0 to 18, with higher scores indicating more severe neurological deficits. Rats were scored by two independent observers blinded to the treatment groups to reduce observer bias. When the results were different, consistent opinions were adopted.

The light-dark box experiment system consisted of a light-dark box (45 cm×27 cm×27 cm), an electrical stimulation controller, a computer, and a behavioral analysis system. The top of the dark box was covered, and the bottom of the dark box was equipped with a copper grid that can conduct electricity. The light box had a lighting device, and there was a partition wall between the light and dark boxes. The partition wall had an opening (7.5 cm×7.5 cm) through which the test rat could move between the two boxes. In the training phase, the rat was placed in the light box. After 10 seconds, the

door between the light and dark boxes was opened. Due to the rat's natural tendency to move toward the dark, they would quickly enter the dark box. Once the rat fully entered the dark box, the door between the two boxes was closed, and a 0.5 mA electrical stimulation was applied for 2 seconds. A test phase was performed on the next day. The rat was placed in the light box again, and the door between the light and dark boxes was opened. The number of transitions between the light and dark compartments reflects the animal's exploratory activity, where lower transition counts indicate higher anxiety levels. A normal exploratory range is typically 5–15 transitions within a 5-minute period, while ≤ 3 transitions indicate a pronounced anxiety phenotype. The duration of time spent in the dark compartment and the latency to first enter the dark zone provide additional measures of anxiety-like behavior.

The MWM was consisted of a circular water tank with a diameter of 1.6 meters and a height of 50 cm, a submerged platform with a diameter of 10 cm and 1 cm below the water surface, and a video tracking system. Four equally spaced points (east, south, west, and north) were marked on the walls of the tank as entry points. The water temperature was maintained at $25 \pm 1°C$. A training phase was started at 8 AM each day, the rats were placed into the water from the four different entry points with positioned facing the tank wall. The video tracking system recorded the rat's movement trajectory and the time taken to reach the platform. When the rat found and climbed onto the platform, it was allowed to rest for 30 seconds before the next trial. If the rat failed to find the platform within 60 seconds, the experimenter gently guided the rat to the platform with a stick and allowed it to rest on the platform for 30 seconds. The trial time was recorded as 60 seconds. Each rat underwent four training trials per day with a 30-minute interval between each trial, for a total of four days. A test phase was performed on the 5th day with the platform removed from the maze. The rat was placed at the midpoint of the wall opposite the platform's original location. The time it took for the rat to first enter the quadrant where the platform was located, as well as the amount of time the rat spent in that quadrant was recorded. The total test time was 60 seconds.

## Histopathological analysis

After the completion of behavioral and neurological assessments, rats were euthanized. Euthanasia was performed via intravenous injection of potassium chloride (KCl, 1–2 mg/kg, 2M solution) administered through the tail vein, and their brains were removed. The brains were fixed in 4% paraformaldehyde and then sectioned coronally at a thickness of 6 μm. Sections from the hippocampal CA1 region were stained with hematoxylin and eosin (H&E).

Nissl staining was used to assess neuronal damage, determine neuron viability, and evaluate tissue necrosis. After deparaffinization and rehydration through graded alcohols, the sections were immersed in a solution of cresyl violet or toluidine blue (0.1–0.5%) for approximately 5–10 minutes. The sections were then briefly rinsed in distilled water before undergoing differentiation in 95% ethanol containing a few drops of acetic acid until the background appeared clear. Finally, the tissue sections were dehydrated through graded alcohols, cleared in xylene, and mounted.

TUNEL staining was used to detect DNA fragmentation associated with apoptotic cells. After deparaffinization and rehydration, the tissue sections were permeabilized with Proteinase K (10–20 μg/mL) for 15–30 minutes at room temperature. Then they were incubated in a TUNEL reaction mixture of terminal deoxynucleotidyl transferase and fluorescein-dUTP for 60 minutes at 37°C in a humidified chamber. For light microscopy visualization, the fluorescent signal was converted using an anti-fluorescein antibody conjugated with peroxidase and DAB substrate. After counterstaining with methyl green or hematoxylin, the sections are dehydrated, cleared, and mounted.

## Transmission electron microscopy

Samples of hippocampal tissue were prepared for transmission electron microscopy. Small pieces of the hippocampus (1–3 mm³) were fixed in 2.5% glutaraldehyde followed by post-fixation in 0.5% osmium tetroxide. The samples were then dehydrated in a graded series of ethanol and embedded in Spurr resin. Ultra-thin sections (70 nm) were cut and stained with uranyl acetate and lead citrate. The sections were examined under a ThermoFisher Talos 120 transmission electron

microscope. Mitochondrial morphology was evaluated, including changes in membrane integrity, cristae structure, and the presence of mitochondrial swelling.

**Measurement of oxidative stress and iron metabolism markers.** Blood samples were collected from the left ventricle immediately after euthanasia. Serum levels of oxidative stress markers, including malondialdehyde (MDA), superoxide dismutase (SOD), and glutathione peroxidase (GSH-Px), were measured using commercially available enzyme-linked immunosorbent assay (ELISA) kits. Additionally, hepcidin and ferroportin (FPN1) were quantified to assess the impact of CCI and HBOT on iron homeostasis.

### Quantitative analysis of histological and ultrastructural data

Quantitative assessments were performed to objectively evaluate neuronal and mitochondrial integrity. Neuronal density in the hippocampal CA1 region was quantified by counting intact neurons in three randomly selected high-power fields (400×) per section per animal. The number of TUNEL-positive cells was quantified in three randomly selected fields (400×) per section per animal. Mitochondrial morphometry was quantified by measuring the average mitochondrial cross-sectional area and cristae integrity

### Western blot analysis

Hippocampal tissue was homogenized in ice-cold RIPA buffer supplemented with protease and phosphatase inhibitors. Protein concentrations were determined using the Bradford assay, and equal amounts of protein (30 μg per sample) were separated by 10% SDS-PAGE and transferred to PVDF membranes. Membranes were blocked with 5% bovine serum albumin (BSA) and incubated overnight at 4°C with primary antibodies against BMP6, Smad1/2, phospho-Smad1/2, ferroportin (FPN1), and GAPDH (loading control). After washing, membranes were incubated with HRP-conjugated secondary antibodies. Protein bands were visualized using enhanced chemiluminescence, and densitometric analysis was performed using ImageJ software to quantify protein expression levels.

### RNA extraction and quantitative real-time PCR (qRT-PCR) of ferroportin

Total RNA was extracted by using the RNA-QuickPurification Kit (ES Science, Shanghai, China). PCR was performed using BeyoFast™ SYBR Green qPCR Mix (2X, Low ROX) kit (Beyotime, Shanghai, China). The threshold cycle(Ct) was determined using the QuantStudio3 (Applied Biosystems, USA). The condition for PCR amplification was initial denaturation at 95 ℃ for 2 min followed by 40 cycles of denaturation at 95 ℃ for 10 sec and annealing/extension at 60 ℃ for 30 sec. β-actin was used as an internal control for gene expression. Detailed methods and the PCR primer sequences are listed in Supplementary Methods (S1 File).

### Statistical analysis

Data are expressed as the mean ± standard deviation. Statistical analysis was performed using R (4.4.2, https://www.r-project.org/). After conducting a normal distribution test, comparisons between groups were made using one-way analysis of variance (ANOVA), followed by Tukey's post hoc test for multiple comparisons. If data did not follow a normal distribution, non-parametric Kruskal-Wallistest was used. A p-value of less than 0.05 was considered statistically significant.

## Results

### Neurological assessment

For mNSS test, the rats in the GM group (CCI rats) displayed significant neurological deficits compared to controls (CT group, $p < 0.001$). Rats treated with HBOT (HO group) showed significantly improved neurological function ($p < 0.001$

compared to GM). The group treated with Normobaric Hyperoxia (NH) also demonstrated improved neurological outcomes (p<0.001 compared to GM), but the effect was less significant than in the HO group (p<0.001).

For the LDB test, the rats in the GM group displayed a significant decrease in the number of entries into the dark box compared to the CT group (p<0.001), indicating severe anxiety-like behavior. Both HO group and NH group exhibited a significant increase in the times compared to the GM group (both p<0.001). However, the HO group entered the dark box significantly more times than the NH group (p=0.018). The latency in the HO group and NH group was significantly shorter compared to the GM group (both p<0.001), and there was a significant difference between the HO group and CT group (p<0.001) (Fig 1).

For the MWM test, the rats in the GM group displayed significantly less time spent in the platform's quadrant compared to the CT group (p<0.001), indicating impaired learning and memory. Compared to the GM group, the HO and NH groups showed significantly increased time spent in the platform's quadrant (p=0.027 and p<0.001), but the HO group spent significantly more time than the NH group (p<0.001). The rats in the GM group displayed a significant increase in latency compared to the CT group (p<0.001). The latency in the HO and NH groups was significantly lower than that in the GM group (p=0.007 and p<0.05), but the HO group had a significantly shorter latency than the NH group (p<0.001) (Fig 1).

## Histopathological analysis

The histopathological analysis is shown in Fig 2. H&E staining showed significant neuronal damage in the hippocampal CA1 region of the GM group. The neurons displayed shrunken cell bodies, hyperchromatic nuclei, and vacuolization of the

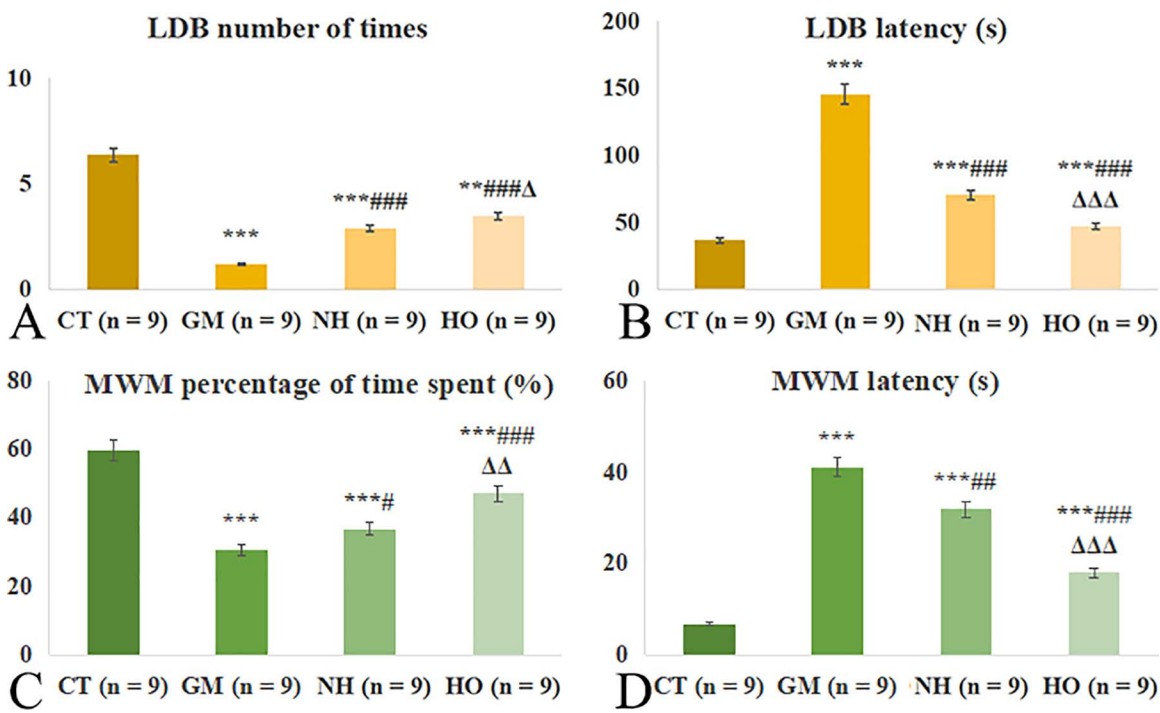

**Fig 1. Effects of hyperbaric oxygen therapy on neurological function of light-dark box (LDB) and Morris water maze (MWM) test.** (A) Number of transitions in the LDB test. (B) Latency time in the LDB test. (C) Percentage of time spent in the target quadrant during MWM test. (D) Latency to find the platform in the MWM test. NH, ischemia-reperfusion model and normobaric hyperoxia group; CT, control group; HO, ischemia-reperfusion model and hyperbaric oxygen therapy group; GM, ischemia-reperfusion model group. *p<0.05, **p<0.01, ***p<0.001 vs. CT group; #p<0.05, ##p<0.01, ###p<0.001 vs. GM group; Δp<0.05, ΔΔp<0.01, ΔΔΔp<0.001 vs. NH group.

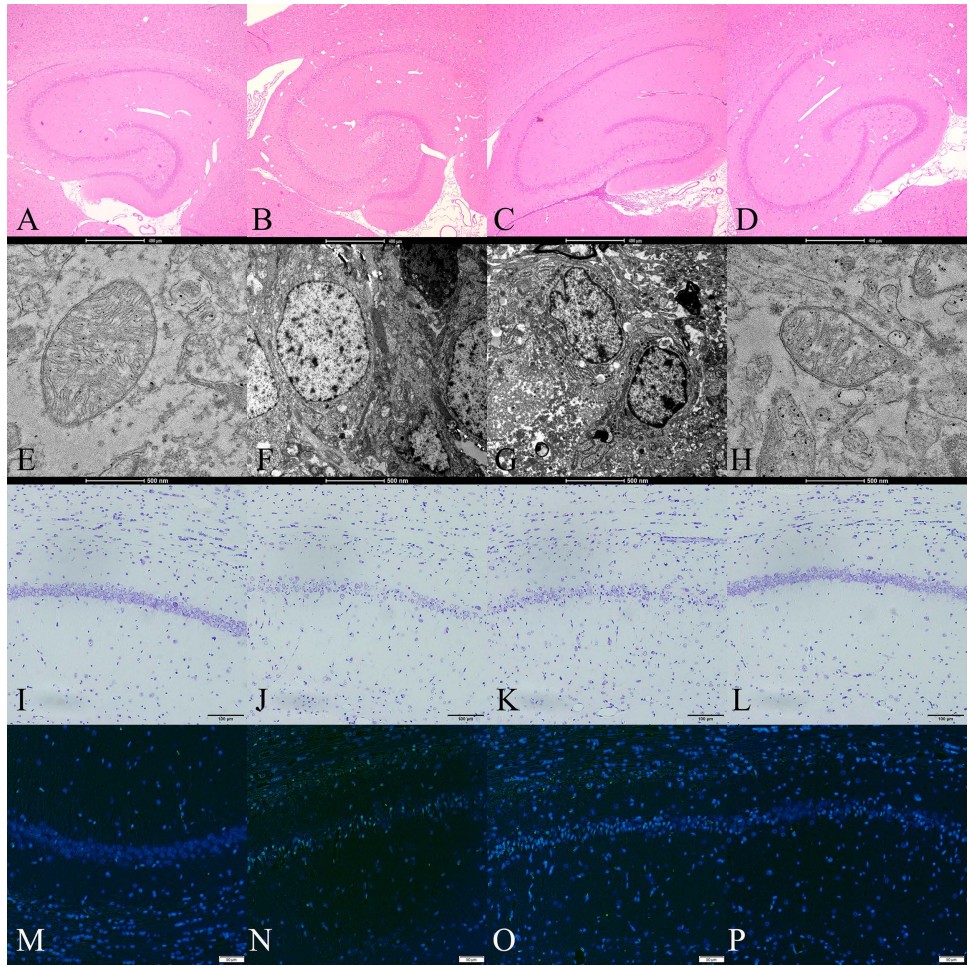

**Fig 2. Histopathological, ultrastructural, nissl, and TUNEL changes in the hippocampal CA1 region of rats after global cerebral ischemia-reperfusion injury.** Representative images showing alterations in the hippocampal CA1 region of rats from different experimental groups: Control (CT, panels A, E, I, M), Ischemia-Reperfusion (GM, panels B, F, J, N), Ischemia-Reperfusion + Normobaric Hyperoxia (NH, panels C, G, K, O), and Ischemia-Reperfusion + Hyperbaric Oxygen Therapy (HO, panels D, H, L, P). A-D: H&E staining showing histopathological changes (Magnification: 400×; Scale bar: 400 μm). (A) Control group displays normal cellular architecture with intact pyramidal neurons. (B) Ischemia-reperfusion group exhibits marked neuronal loss and pyknotic nuclei. (C) Normobaric Hyperoxia shows moderate improvement compared to the ischemia group. (D) HBOT treatment demonstrates notable neuroprotection with preserved cellular morphology. E-H: Transmission electron microscopy of CA1 neurons (Magnification: 10,000×; Scale bar: 500 nm). (E) Control neurons show normal ultrastructure with intact organelles. (F) Ischemia-reperfusion causes severe mitochondrial swelling, endoplasmic reticulum dilation, and nuclear condensation. (G) Normobaric Hyperoxia shows intermediate ultrastructural preservation. (H) HBOT-treated samples display reduced ultrastructural damage with relatively preserved mitochondria. I-L: Nissl staining for neuronal survival assessment (Magnification: 400×; Scale bar: 100 μm). (I) Control section shows abundant Nissl bodies in neuronal cytoplasm. (J) Ischemia-reperfusion leads to significant loss of Nissl substance. (K) Normobaric Hyperoxia shows partial preservation of Nissl substance. (L) HBOT treatment preserves Nissl bodies, indicating improved neuronal viability. M-P: TUNEL assay for apoptosis detection (Magnification: 400×; Scale bar: 50 μm). (M) Control tissue shows minimal TUNEL-positive neurons. (N) Ischemia-reperfusion results in extensive TUNEL-positive neurons, indicating widespread apoptosis. (O) Normobaric Hyperoxia moderately decreases apoptotic neuron death compared to the ischemia group. (P) HBOT treatment significantly reduces TUNEL-positive neurons.

surrounding tissue. In contrast, the HO group exhibited significantly less neuronal damage, with a greater number of intact neurons and reduced vacuolization. The NH group also showed some protective effects, with moderate preservation of neuronal structures compared to the GM group.

Nissl staining revealed that the neurons exhibited substantially reduced Nissl substance, pyknotic nuclei, and decreased neuronal density with irregular cellular arrangements in the hippocampal CA1 region of the GM group. In contrast, the HO group demonstrated markedly preserved neuronal integrity, with abundant Nissl substance in the cytoplasm, well-defined cell boundaries, and organized cellular architecture. The NH group also showed moderate neuroprotective effects compared to the GM group, though some neurons still displayed chromatolysis and structural alterations.

TUNEL staining demonstrated extensive DNA fragmentation in the hippocampal CA1 region of the GM group. The apoptotic cells were characterized by condensed chromatin and fragmented nuclei with strong TUNEL-positive neurons. In contrast, the HO group exhibited significantly reduced TUNEL-positive neurons, with only scattered apoptotic neurons observed. The NH group also showed moderate protective effects against apoptosis, with an intermediate number of TUNEL-positive neurons compared to the GM group.

## Ultrastructural changes in mitochondria

The TEM analysis is shown in Fig 2. Results revealed severe mitochondrial damage in the GM group, characterized by mitochondrial shrinkage, disrupted cristae, and increased membrane density. In the HO group, the structural integrity of mitochondria was notably preserved, with fewer signs of damage such as membrane rupture or cristae loss. The NH group displayed partial protection of mitochondrial structures, but the damage was not significant than the HO group.

**Oxidative stress and iron metabolism markers.** The GM group exhibited significantly elevated levels of MDA compared to the CT group ($p < 0.001$). In contrast, the HO group showed significantly reduced MDA levels compared to the GM group ($p < 0.001$). The NH group also demonstrated reduced MDA levels compared to the GM group, but the reduction was not significant. SOD and GSH-Px levels were significantly lower in the GM group compared to the CT group (both $p < 0.001$). HBOT ($p < 0.001$ for SOD and $p < 0.001$ for GSH-Px) and Normobaric Hyperoxia treatment ($p = 0.052$ for SOD and $p = 0.023$ for GSH-Px) significantly restored these enzyme levels compared to the GM group, while Normobaric Hyperoxia showed moderate restoration.

The GM group showed significantly elevated serum hepcidin levels compared to the CT group ($p < 0.001$). The HO group showed significantly lower hepcidin levels ($p = 0.040$) compared to the GM group, while the NH group ($p = 0.038$) showed a moderate decrease.

The neuronal density in the hippocampal CA1 region was significantly reduced in the GM group compared with the CT group ($p < 0.001$). Both the NH and HO groups exhibited increased neuronal density compared to the GM group ($p < 0.001$ and $p < 0.001$, respectively), with HO showing the greatest preservation. Similarly, the percentage of TUNEL-positive cells was markedly higher in the GM group than in the CT group ($p < 0.001$), while HO significantly reduced apoptotic neuron counts ($p < 0.001$). Mitochondrial morphometric analysis revealed that mitochondrial area and cristae integrity were severely compromised after ischemia-reperfusion but substantially restored in HBOT-treated animals ($p < 0.001$ vs. GM).

## Hepcidin mRNA expression

Quantitative PCR analysis showed a significant upregulation of hepcidin mRNA in the GM group (Fig 3). The HO group treated with HBOT displayed significantly higher hepcidin mRNA levels. The NH group also showed reduced hepcidin mRNA expression compared to the GM group, but the effect was less significant than in the HO group.

Quantitative PCR analysis revealed that hepcidin mRNA expression was markedly elevated in the GM group compared to the control (CT) group ($p < 0.001$). HBOT (GA group) significantly reduced hepcidin mRNA levels relative to the GM group ($p < 0.01$), whereas the NBHO (formerly CA) group showed a moderate but non-significant reduction (Fig 3A-B). These results indicate that HBOT downregulates hepcidin expression following cerebral ischemia-reperfusion injury.

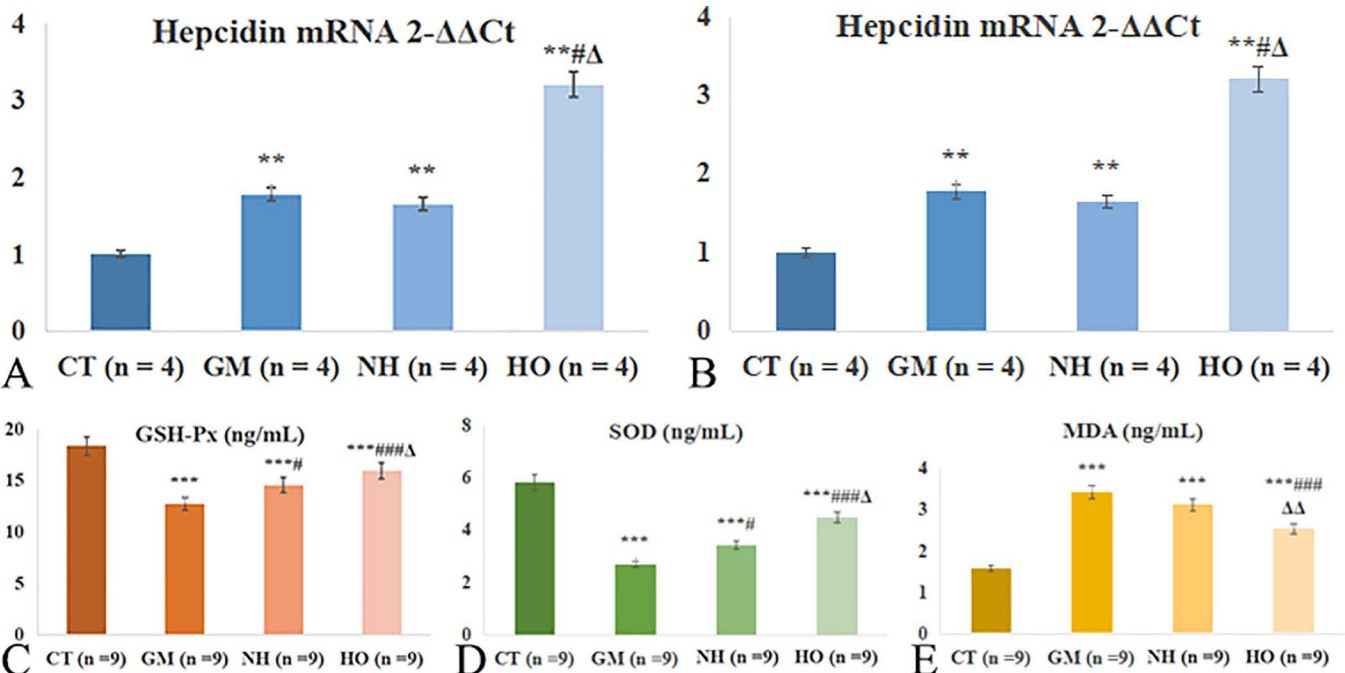

**Fig 3. Effects of hyperbaric oxygen therapy on biochemical parameters in global cerebral ischemia-reperfusion injury.** (A and B) Serum hepcidin protein and mRNA levels measured by enzyme-linked immunosorbent assay (ELISA) and quantitative real-time PCR (qRT-PCR), respectively. (C) Malondialdehyde (MDA) concentrations. (D) Glutathione peroxidase (GSH-Px) activity. (E) Superoxide dismutase (SOD) activity. *$p < 0.05$, **$p < 0.01$, ***$p < 0.001$ vs. CT group; #$p < 0.05$, ##$p < 0.01$, ###$p < 0.001$ vs. GM group; $\Delta p < 0.05$, $\Delta\Delta p < 0.01$, $\Delta\Delta\Delta p < 0.001$ vs. CA group.

## Western blot analysis

In the GM group, a marked increase in BMP6, phospho-Smad1/2, hepcidin and ferroportin expression was shown (Fig 4 and Fig 5). HBOT treatment significantly reduced the expression of BMP6 and phospho-Smad1/2, which was accompanied by decreased expression of hepcidin and ferroportin. The NH group showed partial regulation of these proteins, with less effects compared to the HO group.

## Discussion

The present study investigated the neuroprotective effects of HBOT on CCI rats, focusing on its ability to modulate the BMP6/Smad-Hepcidin pathway and reduce ferroptosis. The findings demonstrate that HBOT significantly improves neurological function, preserves neuronal and mitochondrial integrity, reduces oxidative stress, and regulates iron metabolism. These results provide new insights into the mechanisms underlying the neuroprotective effects of HBOT, highlighting its potential as a therapeutic strategy for CCI.

CCI usually leads to severe neurological deficits due to disrupted cerebral blood flow and subsequent neuronal death. The improvement in neurological function seen in HBOT-treated animals is consistent with earlier reports. Studies demonstrated that HBOT enhances oxygen delivery to hypoxic brain tissue, promoting cellular recovery and improving neurological outcomes [13]. Our results further confirm the beneficial role of HBOT in reducing neurological deficits caused by cerebral ischemia. Mitochondria are highly susceptible to ischemic damage [14]. Previous studies showed that CCI induced cognitive impairment via hippocampal CA1 region damage and the ultrastructural changes objectively in this region after hypoxia [15,16]. Histological analysis revealed that HBOT significantly reduced neuronal damage in the hippocampal CA1 region. The preservation of neuronal structure in HBOT-treated rats aligns with findings from other studies

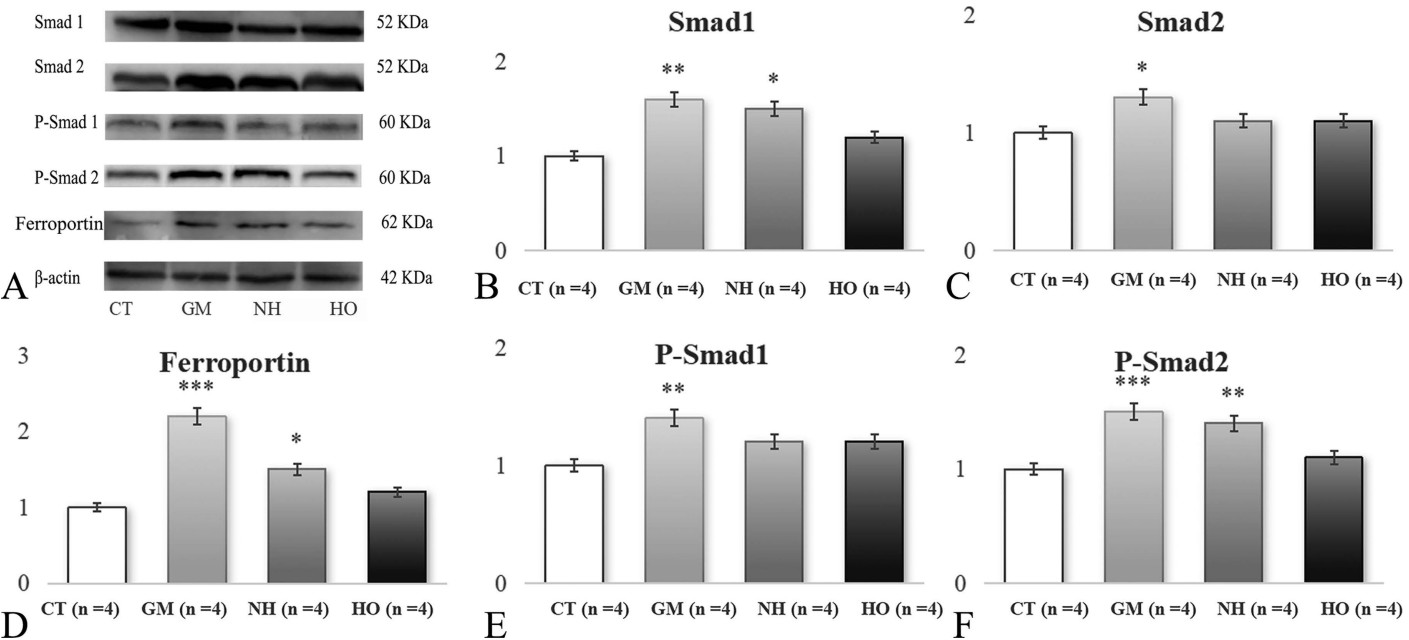

**Fig 4. Western blot analysis of protein expression levels in hippocampal tissues across different experimental groups (n = 4 for each group).**
(A) Western blot images showing protein bands for Smad1 (52 kDa), Smad2 (52 kDa), phosphorylated Smad1 (P-Smad1, 60 kDa), phosphorylated Smad2 (P-Smad2, 60 kDa), Ferroportin/SLC40A1 (62 kDa), and β-actin (42 kDa, loading control). B-F: Quantification of protein expression of (B) Smad1, (C) Smad2, (D) Ferroprotein (SLC40A1), (E) P-Smad1, and (F) P-Smad2. The protein expression levels are normalized to β-actin and presented as folds of control. *$p < 0.05$, **$p < 0.01$, ***$p < 0.001$ vs. CT group; #$p < 0.05$, ##$p < 0.01$, ###$p < 0.001$ vs. GM group; Δ$p < 0.05$, ΔΔ$p < 0.01$, ΔΔΔ$p < 0.001$ vs. CA group.

that reported reduced neuronal apoptosis following HBOT [17]. The results revealed that HBOT preserved mitochondrial integrity, reducing the extent of mitochondrial swelling and cristae disruption.

Previous studies suggested that oxidative stress plays a major role in the pathogenesis of CCI, and mitigating oxidative damage is crucial for improving outcomes [18]. In this study, HBOT significantly reduced serum levels of MDA, a marker of lipid peroxidation, while restoring antioxidant enzyme activities (SOD and GSH-Px). This is consistent with previous reports showing that HBOT increases tissue oxygenation and enhances antioxidant defenses, thus reducing oxidative stress and preventing further damage [19]. The reduction in oxidative stress markers in the HBOT-treated group likely contributes to the observed neuroprotection [20]. Our data show HBOT downregulates BMP6/Smad-Hepcidin, lowers hepcidin, restores ferroportin, and mitigates iron-related oxidative damage. Hepcidin-ferroportin controls intracellular iron, influencing Fenton chemistry, lipoxygenase activity, and GSH/GPX4 defenses. HBOT thus reduces pro-ferroptotic iron, preserves export, lowers MDA, and increases GPX4 expression, forming a multi-layered neuroprotection. These associations are consistent with prior work.

The regulation of iron metabolism is increasingly recognized as an important factor in CCI due to its role in ferroptosis, a form of iron-dependent cell death [21]. Our study demonstrated that CCI upregulated the BMP6/Smad-Hepcidin pathway, leading to elevated hepcidin levels and iron overload and then to ferroptosis. However, HBOT significantly downregulated BMP6/Smad pathway, reduced hepcidin expression and restored ferroportin levels. This suggests that HBOT helps mitigate iron overload and suppress ferroptosis. These findings are consistent with studies that have linked the BMP6/Smad-Hepcidin pathway to the regulation of iron metabolism. By regulating iron homeostasis, BMP6/Smad-Hepcidin pathway limits iron-mediated oxidative damage via complementing antioxidative strategies like GSH and GPX4 enhancement [22]. BMP/Smad signaling shows anti-inflammatory effects by suppressing NF-κB activation [23]. Additionally,

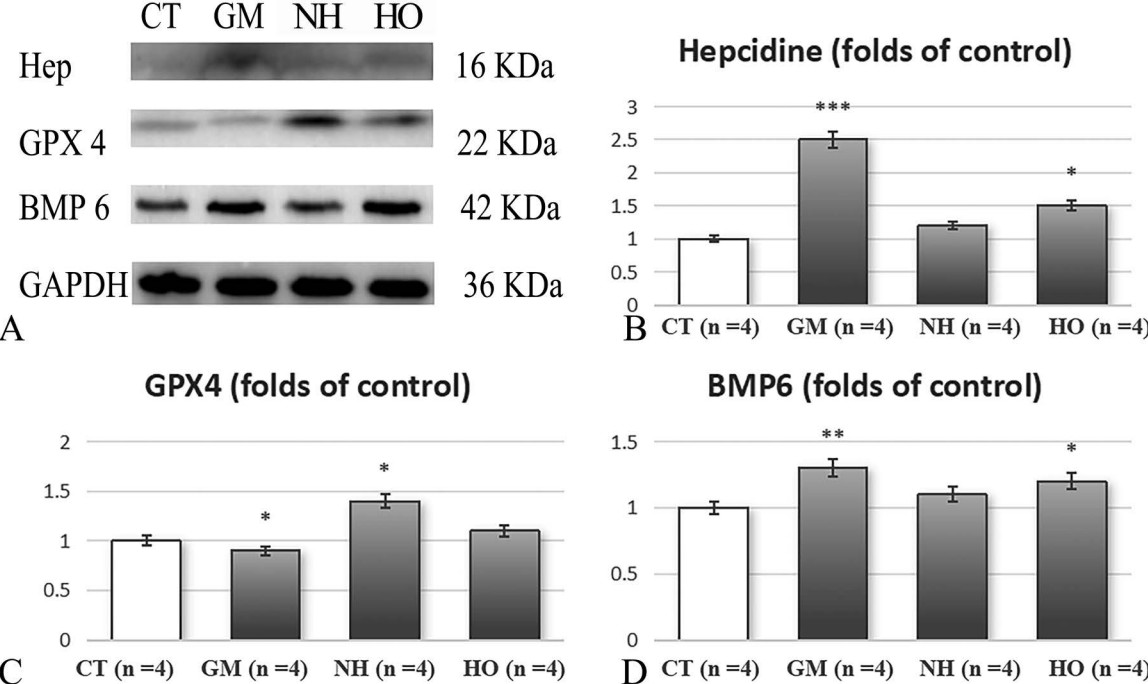

**Fig 5. Western blot analysis of protein expression levels in hippocampal tissues of rats across different experimental groups (n = 4 for each group).** (A) Western blot images showing protein bands for Hepcidin (16 kDa), Glutathione Peroxidase 4 (GPX4, 22 kDa), Bone Morphogenetic Protein 6 (BMP6, 42 kDa), and GAPDH (36 kDa, loading control). B-D: Densitometric quantification of protein expression of (B) Hepcidin, (C) GPX4, and (D) BMP6. The protein expression levels are normalized to GAPDH and presented as folds of control. *$p < 0.05$, **$p < 0.01$, ***$p < 0.001$ vs. CT group; #$p < 0.05$, ##$p < 0.01$, ###$p < 0.001$ vs. GM group; $\Delta p < 0.05$, $\Delta\Delta p < 0.01$, $\Delta\Delta\Delta p < 0.001$ vs. CA group.

BMPs can activate a pro-regenerative transcription program in neurons through the Smad-mediated canonical pathway during axon regeneration [24]. The multifaceted nature of BMP6/Smad-Hepcidin signaling makes it valuable for addressing the complex pathophysiology of CCI. Our findings indicate that HBOT may offer neuroprotection by modulating the BMP6/Smad pathway involved in ferroptosis. Our findings appear to contrast with the canonical role of the BMP6/Smad-Hepcidin pathway, which under physiological conditions limits systemic iron availability and thereby reduces ferroptosis. However, during cerebral ischemia-reperfusion injury, the pathway's regulation becomes tissue- and context-dependent. In the injured brain, excessive activation of BMP6/Smad-Hepcidin signaling may promote intracellular iron retention by suppressing ferroportin-mediated iron export from neurons and glial cells. This pathological accumulation of intracellular iron expands the labile iron pool and enhances lipid peroxidation, ultimately facilitating ferroptosis rather than inhibiting it. Recent studies showed that BMP6/Smad activation has been linked to inflammation and ferroptosis in ischemic stroke models [22]. Suppression of hepcidin signaling could improve neurological recovery [25]. Therefore, the downregulation of BMP6/Smad-Hepcidin signaling by HBOT observed in our study may represent a protective normalization of dysregulated iron handling within neural tissue rather than a simple suppression of physiological iron control.

While our study demonstrates significant neuroprotective effects of HBOT, it is important to note that several limitations should be acknowledged for this study. First, the study was conducted on a small sample size of Wistar rats, which may limit the generalizability of the findings to other species. Second, the study employed a specific protocol for HBOT (2.5 ATA for 120 minutes over five consecutive days), and it remains unclear whether varying the pressure, duration, or timing of therapy might yield different therapeutic results. Future studies should investigate the optimal HBOT parameters for maximal efficacy. Third, while the study focused on the BMP6/Smad-Hepcidin pathway, other molecular mechanisms,

such as inflammation, autophagy, and apoptosis, were not fully explored. A more comprehensive analysis of these pathways would provide a better understanding of the multifaceted protective mechanisms of HBOT. Furthermore, we did not employ pharmacological or genetic manipulations such as BMP6 knockdown or ferrostatin-1 treatment to confirm causality. Future studies incorporating such interventions are needed to definitively establish the mechanistic role of this pathway in HBOT-mediated neuroprotection. In addition, the study was limited to short-term outcomes, and the long-term effects of HBOT on neurological recovery and overall brain function were not assessed. Future research should include longer follow-up periods to evaluate the sustainability of HBOT's protective effects. Last, rodent brains differ markedly from humans in size, white/gray matter distribution, cerebrovascular anatomy, and metabolism, limiting direct extrapolation. Our global ischemia model does not fully capture human focal strokes, age/comorbidity effects, or clinical HBOT protocols. Future work should leverage larger animal models, aged/diabetic animals, varied HBOT parameters, longer follow-up, and early-phase clinical trials to bridge these gaps, while focusing on mechanistic biomarkers.

In conclusion, this study showed that HBOT significantly improves neurological function, reduces neuronal damage, and regulates iron metabolism in a rat model of CCI. The modulation of the BMP6/Smad-Hepcidin pathway and the suppression of ferroptosis were shown as the mechanisms underlying the neuroprotective effects of HBOT. These findings highlight the potential of HBOT as a therapeutic strategy for CCI and underscore the need for further clinical studies to explore its application in patients suffering from CCI.

## Supporting information

**S1 File. Supplementary Methods.**
(DOCX)

**S2 File. ARRIVE guidelines 2.0 – English.**
(PDF)

**S1 Images. S1 raw images.**
(PDF)

**S1 Data. Rawdata.**
(ZIP)

## Author contributions

**Conceptualization:** Xiu-Ju Li.

**Data curation:** Lan-Zhao Wang, Ji-Hong Zhang, Ji-Min Shi, Xiu-Ju Li.

**Formal analysis:** Xiu-Ju Li.

**Funding acquisition:** Lan-Zhao Wang.

**Supervision:** Xiu-Ju Li.

**Writing – original draft:** Lan-Zhao Wang, Ji-Hong Zhang, Ji-Min Shi.

**Writing – review & editing:** Xiu-Ju Li.

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
