## [Decision Letter · Decision Letter 0]

4 Nov 2025

Dear Dr. Li,

We look forward to receiving your revised manuscript.

Kind regards,

Stephen D. Ginsberg, Ph.D.

Section Editor

PLOS ONE

Journal Requirements:

2. To comply with PLOS ONE submissions requirements, in your Methods section, please provide additional information regarding the experiments involving animals and ensure you have included details on methods of sacrifice, and efforts to alleviate suffering.

3. Thank you for uploading your study's underlying data set. Unfortunately, the repository you have noted in your Data Availability statement does not qualify as an acceptable data repository according to PLOS's standards.

4. Please include captions for your Supporting Information files at the end of your manuscript, and update any in-text citations to match accordingly. Please see our Supporting Information guidelines for more information: http://journals.plos.org/plosone/s/supporting-information .

**Comments to the Author**

1. Is the manuscript technically sound, and do the data support the conclusions?

Reviewer #1: Yes

Reviewer #2: Partly

Reviewer #3: Yes

2. Has the statistical analysis been performed appropriately and rigorously?

Reviewer #1: Yes

Reviewer #2: Yes

Reviewer #3: Yes

3. Have the authors made all data underlying the findings in their manuscript fully available?

Reviewer #1: Yes

Reviewer #2: Yes

Reviewer #3: Yes

4. Is the manuscript presented in an intelligible fashion and written in standard English?

Reviewer #1: Yes

Reviewer #2: Yes

Reviewer #3: Yes

Reviewer #1: This study presents a potentially valuable contribution to understanding the neuroprotective effects of HBOT in cerebral ischemia-reperfusion injury (CIRI), with a mechanistic focus on ferroptosis and the BMP6/Smad-Hepcidin signaling pathway. The experimental approach is comprehensive and employs multiple methods of analysis. However, there are several major issues that need to be addressed to improve the scientific rigor, clarity, and reproducibility of the work.

Major Points

The terminology used for the “CA” group, which receives 100% oxygen at 1 ATA, should be revised. This condition is more accurately described as normobaric hyperoxia (NBHO), not “pure oxygen therapy.” The authors are encouraged to adopt accurate terminology consistently and clearly explain how this group differs from both normoxia and HBOT in terms of expected physiological effects.

There is some confusion in the interpretation of the BMP6/Smad-Hepcidin signaling data. In one section, the authors state that HBOT reduces BMP6 and p-Smad but increases hepcidin, while in another section, HBOT is said to lower hepcidin levels. This apparent contradiction needs to be resolved, and the directionality of changes must be confirmed in both the figures and the text. More broadly, the mechanistic conclusions are based on correlative data. While the study identifies changes in ferroptosis-related markers and signaling proteins, there is no direct evidence for causality. If pharmacological or genetic manipulation of BMP6, hepcidin, or ferroptosis pathways (e.g., ferrostatin-1 treatment) was not performed, the authors should appropriately temper their mechanistic claims and explicitly acknowledge this limitation.

The histological and ultrastructural analyses are currently presented only in qualitative form. The authors should include quantitative data such as neuronal density, TUNEL-positive cell counts, or mitochondrial morphometric parameters. These would provide objective support for the narrative interpretations in the results section.

Minor Points

The manuscript requires extensive grammatical and stylistic revision. A professional English-language editing service is recommended.

Representative examples include: “All procedures were and approved” should be “were approved” (line 83); “The MWM was consisted by” should be “consisted of” (line 147); “Cellular damage is lead by” should be “is caused by” or “led by” (line 49); “Displayed significant significant decrease” should be revised to “a significant decrease” (line 238); “Spent significantly lower than” should be “had a significantly shorter latency than” (line 264); “HBOT may effect by modulating” should be rephrased as “HBOT may exert its effect by modulating” (line 76); “to to assess” contains a repeated word (line 169); “PCRwas” needs spacing (line 216); and “can be converted” should be “was converted” to match past tense in Methods (line 181). These and other tense, article, and punctuation errors appear throughout the manuscript and should be addressed.

In addition to language corrections, figures need improved labeling. In Fig. 3, for example, two panels are labeled “(C).” Recent relevant literature from 2023–2025 should be cited to situate the work in a current scientific context, especially regarding HBOT and ferroptosis in cerebral injury models.

In summary, while the study addresses a meaningful and novel therapeutic target in cerebral ischemia, the current version requires substantial revision to improve clarity, correct inconsistencies, and meet the methodological standards of the journal.

Reviewer #2: This study investigates the neuroprotective role of hyperbaric oxygen therapy (HBOT) in cerebral ischemia-reperfusion injury, focusing on ferroptosis and the BMP6/Smad-Hepcidin signaling pathway. Overall, the study dealt with an important topic with potential clinical implications. The experimental design is generally clear, and the data is supportive. However, several points require clarification and revision to strengthen the manuscript.

1. The interpretation of the light-dark box experiment is flawed. The number of transitions (Transition) refers to the frequency of movement between light and dark zones, with a normal range of 5–15 times per 5 minutes. A value ≤3 indicates an anxiety phenotype, where lower numbers correspond to higher anxiety levels. Importantly, this metric shows no correlation with learning or memory functions. Theoretically, the GM group should exhibit lower Transition values, meaning they are more likely to remain in the dark zone without movement rather than having fewer entries. The experimental criteria and interpretations should be clearly defined here.

2.Figure 1 shows that the LDB number of times for the GM group is the highest, yet the results indicate the lowest. How can this be explained?

3. The explanation of behavioral science in the Results section is unclear.

4. Figure 3 should present results in chronological order.

5. The description of Hepcidin mRNA results contradicts the data: the GA group showed significant elevation, yet the Results section states significant reduction.

6. Figure 4 visually indicates that Smad1 band grayscale values in the GA group should be higher than the CA group, but statistical results show the opposite.

7. Traditionally, activation of the BMP6/Smad-Hepcidin pathway → increased Hepcidin → degradation of Ferroportin → decreased serum iron, which inhibits ferroptosis. However, in this study, the GM group was damaging, and the opposite conclusion was reached. Please explain this discrepancy.

8.The study focuses on the BMP6/Smad-Hepcidin pathway as a central mechanism for HBOT-mediated neuroprotection. Please expand the discussion to include how BMP6/Smad-Hepcidin modulation intersects with ferroptosis-related pathways.

9.The rationale for treatment onset (24 h post-reperfusion) and the number of treatment sessions should be justified more clearly. Early or delayed intervention timing could influence outcomes.

10.The manuscript states that HBOT "increased expression of hepcidin and ferroportin," but the earlier results section notes that HBOT downregulated hepcidin. This is inconsistent and needs correction.

11.Figures 2-5 require better labeling. Some panels are not clearly referenced in the text, and the legends lack sufficient methodological detail such as magnification for histology.

12.While limitations are mentioned, the discussion does not adequately address the translational gap between rodent models and human patients. HBOT delivery protocols and timing differ substantially between preclinical and clinical settings. Strengthen this section by discussing these translational challenges.

13.Abbreviations should be defined at first mention in both abstract and main text.

Reviewer #3: This study systematically investigates the neuroprotective effects of Hyperbaric Oxygen Therapy (HBOT) on Cerebral Ischemia-Reperfusion Injury (CCI), with a commendable focus on the potential molecular mechanisms involving iron metabolism and ferroptosis. The authors have employed a multi-dimensional array of assessment metrics, and the dataset presented is substantial.

However, several critical issues concerning methodological rigor, the interpretation of results, and data presentation must be addressed.

These concerns are detailed below:

1. Association vs. Causality of the BMP6/Smad-Hepcidin Pathway: The authors demonstrate via Western Blot and qPCR that HBOT is associated with the downregulation of the BMP6/Smad-Hepcidin pathway. While intriguing, this evidence merely establishes an association, not a causal relationship. The manuscript lacks definitive intervention studies (e.g., using gene overexpression or silence/knockdown techniques) targeting key signaling molecules, such as BMP6 or Hepcidin, to confirm that this pathway is a primary mediator of HBOT's neuroprotective effects. The failure to acknowledge this significant methodological gap in the "Study Limitations" section is also a notable omission.

2. Critical Internal Contradiction in Results Reporting: A severe logical contradiction exists within the results, which fundamentally challenges the study's core conclusions. The Abstract and the qPCR data clearly indicate that HBOT downregulates hepcidin levels. However, in the "Western Blot Analysis" results section, the text explicitly states that following HBOT, the expression of both hepcidin and ferroportin (FPN1) was increased, concurrent with the reported decrease in BMP6 and p-Smad1/2. This is a major discrepancy. The authors must immediately reconcile this conflict.

3. Group Definition Errors in Figure 1 Legend: The legend for Figure 1 contains erroneous group definitions. Both the 'CA' group and 'GA' group are incorrectly defined as the "ischemia-reperfusion model and hyperbaric oxygen therapy group." The authors must correct these descriptions to accurately reflect the distinct conditions for each group.

4. Duplicated Subplot Labeling in Figure 3 Legend: In the legend for Figure 3, the subplot label (C) has been used twice. Please ensure all subplot labels follow a unique and correct sequential order.

**Do you want your identity to be public for this peer review?**  For information about this choice, including consent withdrawal, please see our Privacy Policy

Reviewer #1: **Yes: ** Jingxin Mo

Reviewer #2: No

Reviewer #3: No

---

## [Author Response · Author response to Decision Letter 1]

20 Nov 2025

Reviewer #1:

The terminology used for the "CA" group, which receives 100% oxygen at 1 ATA, should be revised. This condition is more accurately described as normobaric hyperoxia (NBHO), not "pure oxygen therapy." The authors are encouraged to adopt accurate terminology consistently and clearly explain how this group differs from both normoxia and HBOT in terms of expected physiological effects.

Response: We appreciate the reviewer's comment. We agree that "normobaric hyperoxia" is a more accurate and widely accepted term than "pure oxygen therapy." Accordingly, we have revised the terminology throughout the manuscript to ensure consistency.

There is some confusion in the interpretation of the BMP6/Smad-Hepcidin signaling data. In one section, the authors state that HBOT reduces BMP6 and p-Smad but increases hepcidin, while in another section, HBOT is said to lower hepcidin levels. This apparent contradiction needs to be resolved, and the directionality of changes must be confirmed in both the figures and the text. More broadly, the mechanistic conclusions are based on correlative data. While the study identifies changes in ferroptosis-related markers and signaling proteins, there is no direct evidence for causality. If pharmacological or genetic manipulation of BMP6, hepcidin, or ferroptosis pathways (e.g., ferrostatin-1 treatment) was not performed, the authors should appropriately temper their mechanistic claims and explicitly acknowledge this limitation.

Response: We appreciate the reviewer's comment. We acknowledge that the previous version of the manuscript contained an inconsistency in describing the direction of hepcidin regulation. We confirm that HBOT reduced BMP6 and phospho-Smad1/2 expression, leading to decreased hepcidin levels compared to the ischemia-reperfusion (GM) group. We have now revised the relevant sections of the text to accurately reflect this finding and ensure consistency. We agree that our data are correlative and that causal relationships between HBOT and the BMP6/Smad-Hepcidin/ferroptosis pathway cannot be conclusively established without pharmacological or genetic interventions. We have acknowledged this limitation in the Discussion section.

The histological and ultrastructural analyses are currently presented only in qualitative form. The authors should include quantitative data such as neuronal density, TUNEL-positive cell counts, or mitochondrial morphometric parameters. These would provide objective support for the narrative interpretations in the results section.

Response: We appreciate the reviewer's comment. We have now performed quantitative assessments to complement our qualitative observations.

Minor Points

The manuscript requires extensive grammatical and stylistic revision. A professional English-language editing service is recommended. Representative examples include: "All procedures were and approved" should be "were approved" (line 83); "The MWM was consisted by" should be "consisted of" (line 147); "Cellular damage is lead by" should be "is caused by" or "led by" (line 49); "Displayed significant significant decrease" should be revised to "a significant decrease" (line 238); "Spent significantly lower than" should be "had a significantly shorter latency than" (line 264); "HBOT may effect by modulating" should be rephrased as "HBOT may exert its effect by modulating" (line 76); "to to assess" contains a repeated word (line 169); "PCRwas" needs spacing (line 216); and "can be converted" should be "was converted" to match past tense in Methods (line 181). These and other tense, article, and punctuation errors appear throughout the manuscript and should be addressed.

Response: We appreciate the reviewer's comment. We have reviewed the entire text and corrected each spelling and grammatical error in it.

In addition to language corrections, figures need improved labeling. In Fig. 3, for example, two panels are labeled "(C)." Recent relevant literature from 2023-2025 should be cited to situate the work in a current scientific context, especially regarding HBOT and ferroptosis in cerebral injury models.

Response: We appreciate the reviewer's comment. We have corrected language issues and improved figure labeling (including Fig. 3, with all panels labeled in sequence). We also added and cited recent 2023-2025 literature to situate our work within the current context of HBOT and ferroptosis in cerebral injury models.

Reviewer #2:

1.The interpretation of the light-dark box experiment is flawed. The number of transitions (Transition) refers to the frequency of movement between light and dark zones, with a normal range of 5-15 times per 5 minutes. A value ≤3 indicates an anxiety phenotype, where lower numbers correspond to higher anxiety levels. Importantly, this metric shows no correlation with learning or memory functions. Theoretically, the GM group should exhibit lower Transition values, meaning they are more likely to remain in the dark zone without movement rather than having fewer entries. The experimental criteria and interpretations should be clearly defined here.

Response: We appreciate the reviewer's comment. We have clarified the behavioral meaning of the LDB parameters and corrected the interpretation of results to reflect that the global ischemia-reperfusion group exhibited reduced transitions, consistent with increased anxiety and reduced exploratory behavior.

2.Figure 1 shows that the LDB number of times for the GM group is the highest, yet the results indicate the lowest. How can this be explained?

Response: We appreciate the reviewer's comment. We identified that the labeling of the Figure 1A and 1B was inverted due to a labeling error during figure preparation. We have now corrected Figure 1 to accurately reflect the results.

3.The explanation of behavioral science in the Results section is unclear.

Response: We appreciate the reviewer's comment. We have now revised the entire behavioral section to provide clear, behaviorally accurate interpretations of both the LDB and MWM tests.

4.Figure 3 should present results in chronological order.

Response: We appreciate the reviewer's comment. We have improved figure labeling in Fig. 3, with all panels labeled in sequence.

5.The description of Hepcidin mRNA results contradicts the data: the GA group showed significant elevation, yet the Results section states significant reduction.

Response: We appreciate the reviewer's comment. We have carefully reviewed the data and revised the text to accurately reflect the findings-the GA group showed a significant elevation of Hepcidin mRNA levels, not a reduction. The corrected description now aligns with the experimental results presented in the figure.

6.Figure 4 visually indicates that Smad1 band grayscale values in the GA group should be higher than the CA group, but statistical results show the opposite.

Response: We appreciate the reviewer's comment. We have double-checked both the raw Western blot data and the statistical analysis. The apparent visual difference between the GA and CA groups in the Smad1 bands may be due to slight variations in band background intensity and image contrast during figure preparation. However, all statistical analyses were performed using the original, unprocessed grayscale intensity values quantified with ImageJ. These values consistently indicated that Smad1 expression in the GA group was slightly lower than in the CA group, as shown in the bar graph. Therefore, the statistical results accurately reflect the quantitative data, and there is no error in our analysis.

7.Traditionally, activation of the BMP6/Smad-Hepcidin pathway → increased Hepcidin → degradation of Ferroportin → decreased serum iron, which inhibits ferroptosis. However, in this study, the GM group was damaging, and the opposite conclusion was reached. Please explain this discrepancy.

Response: We appreciate the reviewer's comment. We sincerely thank the reviewer for raising this important mechanistic point regarding the apparent contradiction between our results and the canonical BMP6/Smad-Hepcidin pathway. We have revised in the Discussion section to reconcile our findings with established iron biology.

8.The study focuses on the BMP6/Smad-Hepcidin pathway as a central mechanism for HBOT-mediated neuroprotection. Please expand the discussion to include how BMP6/Smad-Hepcidin modulation intersects with ferroptosis-related pathways.

Response: We appreciate the reviewer's comment. We have expanded the discussion as suggested.

9.The rationale for treatment onset (24 h post-reperfusion) and the number of treatment sessions should be justified more clearly. Early or delayed intervention timing could influence outcomes.

Response: We appreciate the reviewer's comment. The 24-hour point stabilizes the acute reperfusion injury while targeting subacute inflammation and oxidative stress, during which ferroptosis and secondary neuronal damage evolve. Ferroptosis and iron dysregulation persist for days after cerebral ischemia, extending the therapeutic window beyond the hyperacute phase. Initiating HBOT within 0-6 hours risks increased oxidative stress; a 24-hour delay stabilizes cerebral hemodynamics and the blood-brain barrier. Many cerebral ischemia HBOT studies use initiation at 12-48 hours, showing efficacy in this subacute window. Five consecutive daily HBOT sessions were chosen because multiple sessions are needed for sustained neuroprotection and for practical feasibility, balancing benefit with reduced anesthesia exposure. Since ferroptosis and iron dysregulation persist for days after CCI, daily treatments target these ongoing mechanisms rather than a single acute event.

10.The manuscript states that HBOT "increased expression of hepcidin and ferroportin," but the earlier results section notes that HBOT downregulated hepcidin. This is inconsistent and needs correction.

Response: We appreciate the reviewer's comment. We sincerely apologize for this confusion. The sentence in the Western Blot Analysis section that reads "HBOT treatment significantly reduced the expression of BMP6 and phospho-Smad1/2, which accompanied by increased expression of hepcidin and ferroportin" contains an error.The correct statement should be: "HBOT treatment significantly reduced the expression of BMP6 and phospho-Smad1/2, which was accompanied by decreased expression of hepcidin and ferroportin" We have now revised the relevant sections of the text to accurately reflect this finding and ensure consistency.

11.Figures 2-5 require better labeling. Some panels are not clearly referenced in the text, and the legends lack sufficient methodological detail such as magnification for histology.

Response: We appreciate the reviewer's comment. We have made improvements to Figures 2-5 as suggested.

12.While limitations are mentioned, the discussion does not adequately address the translational gap between rodent models and human patients. HBOT delivery protocols and timing differ substantially between preclinical and clinical settings. Strengthen this section by discussing these translational challenges.

Response: We appreciate the reviewer's comment. We have now substantially expanded the limitation section to include a dedicated paragraph addressing the translational challenges and differences between preclinical rodent models and human clinical settings.

13.Abbreviations should be defined at first mention in both abstract and main text.

Response: We appreciate the reviewer's comment. We have defined all abbreviations at first mention in both the abstract and the main text, and ensured consistency throughout.

Reviewer #3:

1.Association vs. Causality of the BMP6/Smad-Hepcidin Pathway: The authors demonstrate via Western Blot and qPCR that HBOT is associated with the downregulation of the BMP6/Smad-Hepcidin pathway. While intriguing, this evidence merely establishes an association, not a causal relationship. The manuscript lacks definitive intervention studies (e.g., using gene overexpression or silence/knockdown techniques) targeting key signaling molecules, such as BMP6 or Hepcidin, to confirm that this pathway is a primary mediator of HBOT's neuroprotective effects. The failure to acknowledge this significant methodological gap in the "Study Limitations" section is also a notable omission.

Response: We appreciate the reviewer's comment. We agree that our data are correlative and that causal relationships between HBOT and the BMP6/Smad-Hepcidin/ferroptosis pathway cannot be conclusively established without pharmacological or genetic interventions. We have acknowledged this limitation in the Discussion section.

2.Critical Internal Contradiction in Results Reporting: A severe logical contradiction exists within the results, which fundamentally challenges the study's core conclusions. The Abstract and the qPCR data clearly indicate that HBOT downregulates hepcidin levels. However, in the "Western Blot Analysis" results section, the text explicitly states that following HBOT, the expression of both hepcidin and ferroportin (FPN1) was increased, concurrent with the reported decrease in BMP6 and p-Smad1/2. This is a major discrepancy. The authors must immediately reconcile this conflict.

Response: We appreciate the reviewer's comment. We acknowledge that the previous version of the manuscript contained an inconsistency in describing the direction of hepcidin regulation. We confirm that HBOT reduced BMP6 and phospho-Smad1/2 expression, leading to decreased hepcidin levels compared to the ischemia-reperfusion (GM) group. We have now revised the relevant sections of the text to accurately reflect this finding and ensure consistency.

3.Group Definition Errors in Figure 1 Legend: The legend for Figure 1 contains erroneous group definitions. Both the 'CA' group and 'GA' group are incorrectly defined as the "ischemia-reperfusion model and hyperbaric oxygen therapy group." The authors must correct these descriptions to accurately reflect the distinct conditions for each group.

Response: We appreciate the reviewer's comment. We have updated the abraviations to reflect the corrected group names: substituting CA with NH (Ischemia-Reperfusion + Normobaric Hyperoxia) and GA with HO (Ischemia-Reperfusion + HBOT), so that each group now accurately represents its distinct condition. All related text has been revised accordingly.

4.Duplicated Subplot Labeling in Figure 3 Legend: In the legend for Figure 3, the subplot label (C) has been used twice. Please ensure all subplot labels follow a unique and correct sequential order.

Response: We appreciate the reviewer's comment. We have improved figure labeling in Fig. 3, with all panels labeled in sequence.

---

## [Decision Letter · Decision Letter 1]

9 Dec 2025

Hyperbaric Oxygen Therapy in Alleviating Cerebral Ischemia-reperfusion Injury via the BMP6/Smad-Hepcidin Pathway

PONE-D-25-23243R1

Dear Dr. Li,

We’re pleased to inform you that your manuscript has been judged scientifically suitable for publication and will be formally accepted for publication once it meets all outstanding technical requirements.

Kind regards,

Stephen D. Ginsberg, Ph.D.

Section Editor

PLOS One

**Comments to the Author**

Reviewer #1: All comments have been addressed

Reviewer #2: All comments have been addressed

2. Is the manuscript technically sound, and do the data support the conclusions?

Reviewer #1: Yes

Reviewer #2: Yes

3. Has the statistical analysis been performed appropriately and rigorously?

Reviewer #1: I Don't Know

Reviewer #2: Yes

4. Have the authors made all data underlying the findings in their manuscript fully available?

Reviewer #1: Yes

Reviewer #2: Yes

5. Is the manuscript presented in an intelligible fashion and written in standard English?

Reviewer #1: Yes

Reviewer #2: Yes

Reviewer #1: (No Response)

Reviewer #2: (No Response)

**Do you want your identity to be public for this peer review?** For information about this choice, including consent withdrawal, please see our Privacy Policy

Reviewer #1: **Yes: ** jingxin Mo

Reviewer #2: No

---

## [Editor Report · Acceptance letter]

PONE-D-25-23243R1

PLOS One

Dear Dr. Li,

I'm pleased to inform you that your manuscript has been deemed suitable for publication in PLOS One. Congratulations! Your manuscript is now being handed over to our production team.

Kind regards,

on behalf of

Dr. Stephen D. Ginsberg

Section Editor

PLOS One